# ELO-EVOLVE: A CO-EVOLUTIONARY FRAMEWORK FOR LANGUAGE MODEL ALIGNMENT

## ABSTRACT

Current alignment methods for Large Language Models (LLMs) rely on compressing vast amounts of human preference data into static, absolute reward functions, leading to data scarcity, noise sensitivity, and training instability. We introduce **Elo-Evolve**, a co-evolutionary framework that redefines alignment as dynamic multi-agent competition within an adaptive opponent pool. Our approach makes two key innovations: (1) eliminating Bradley-Terry model dependencies by learning directly from binary win/loss outcomes in pairwise competitions, and (2) implementing Elo-orchestrated opponent selection that provides automatic curriculum learning through temperature-controlled sampling. We ground our approach in PAC learning theory, demonstrating that pairwise comparison achieves superior sample complexity ($O(1/\varepsilon)$ vs $O(1/\varepsilon^2)$) and empirically validate a $4.5\times$ noise reduction compared to absolute scoring approaches. Experimentally, we train a Qwen2.5-7B model using our framework with opponents including Qwen2.5-14B, Qwen2.5-32B, and Qwen3-8B models. Results demonstrate a clear performance hierarchy: point-based methods $<$ static pairwise training $<$ Elo-Evolve across Alpaca Eval 2.0 and MT-Bench, validating the progressive benefits of pairwise comparison and dynamic opponent selection for LLM alignment.

## 1 INTRODUCTION

Current alignment methods for Large Language Models (LLMs) predominantly rely on a two-stage process that compresses vast amounts of human preference data into a static, absolute reward function (Christiano et al., 2017; Ziegler et al., 2019; Ouyang et al., 2022). This paradigm begins by training a reward model to distill human preferences into scalar scores, then optimizes a policy via reinforcement learning to maximize these absolute reward signals. While this approach has proven effective, it suffers from several fundamental limitations that create bottlenecks in scalability and performance.

These limitations manifest in three key areas that constrain alignment effectiveness. First, training effective reward models requires vast amounts of high-quality preference data, which is expensive and difficult to collect at scale, often resulting in poor generalization and reward hacking behaviors. Second, the Bradley-Terry (BT) model commonly used for preference modeling suffers from suboptimal sample complexity and high sensitivity to label noise (Bradley & Terry, 1952; Sun et al., 2024), propagating low-fidelity signals throughout the training process. Third, static reward models struggle to provide discriminative feedback as policies improve, creating optimization challenges in advanced training stages (Stiennon et al., 2020).

To address these limitations, we introduce Elo-Evolve, a dynamic multi-agent competition framework that redefines alignment by eliminating BT model dependencies and learning directly from competitive interactions. Rather than compressing preferences into static absolute scores, our approach maintains an adaptive opponent pool where policies learn through real-time pairwise comparisons against strategically selected opponents.

Our framework makes two key innovations: (1) Direct competitive learning that eliminates the need for static reward model intermediaries by learning directly from win/loss outcomes in pairwise competitions; and (2) Elo-orchestrated opponent selection that implements adaptive curriculum learning,

automatically adjusting training difficulty as the policy evolves by maintaining a pool of opponents tracked through Elo ratings (Elo, 1961).

Elo-Evolve builds upon Group Relative Policy Optimization (GRPO) (Shao et al., 2024), replacing static training data with a dynamic competitive environment. The policy learns by competing against opponents selected through a temperature-controlled Elo-based sampling strategy that ensures optimal training challenge—initially facing similar-strength opponents, then gradually transitioning to stronger challengers as its rating improves.

This competitive framework offers several theoretical and practical advantages. By eliminating the BT bottleneck, our method avoids the sample complexity and noise sensitivity issues inherent in absolute reward modeling. Theoretical results suggest that pairwise comparisons offer superior sample efficiency and noise resilience compared to absolute scoring approaches. The binary nature of competitive outcomes provides consistently strong training signals, while the dynamic opponent selection ensures that training challenge scales appropriately with the policy's evolving capabilities.

Our contributions are threefold: (1) We introduce a co-evolutionary alignment framework that replaces static reward modeling with dynamic multi-agent competition, eliminating both BT model dependencies and the need for explicit reward model training by leveraging LLM judges directly; (2) We develop an Elo-based opponent selection mechanism that implements automatic curriculum learning through temperature-controlled sampling; and (3) We provide empirical validation showing progressive improvements from point-based methods to pairwise comparison to Elo-orchestrated selection across multiple benchmarks. Through extensive experiments on Alpaca Eval and MT-Bench, we demonstrate a clear performance hierarchy: traditional point-based reward methods $<$ static pairwise comparison $<$ Elo-Evolve's dynamic opponent selection, validating both the benefits of competitive learning and adaptive curriculum design. Our framework opens new directions for scalable alignment that bypasses reward model training entirely while providing strong reinforcement signals through direct competitive evaluation.

## 2 THE ELO-EVOLVE FRAMEWORK

### 2.1 FROM STATIC REWARD MODELS TO DYNAMIC COMPETITION

Traditional RLHF can be formalized as optimizing:

$$\pi^* = \arg\max_{\pi} \mathbb{E}_{x \sim \mathcal{D}, y \sim \pi(\cdot|x)}[r_\theta(x, y)] \tag{1}$$

where $\mathcal{D}$ is the prompt distribution and $r_\theta$ is a fixed reward model. The policy is updated by maximizing the scalar score predicted by this reward model.

This objective suffers from the limitations discussed above. Elo-Evolve addresses these issues by reframing alignment as competitive learning within a dynamic multi-agent environment: instead of predicting an absolute score, the policy is directly compared to opponents on the same prompt and learns from its win rate.

**Definition 1 (Co-evolutionary Objective).** Given a competitive environment $\mathcal{E} = \{\pi\} \cup \mathcal{M}$ where $\mathcal{M} = \{M_1, M_2, ..., M_K\}$ is a set of opponent models, the co-evolutionary objective is:

$$\pi^* = \arg\max_{\pi} \mathbb{E}_{M \sim p(M|\pi)}[P(\pi(x) \succ M(x))] \tag{2}$$

where $p(M|\pi)$ is an adaptive opponent sampling distribution orchestrated by the Elo system, and $P(\pi(x) \succ M(x))$ represents the probability that the policy's response to prompt $x$ is preferred over the opponent's response.

### 2.2 ELO-ORCHESTRATED OPPONENT SELECTION

The Elo rating system serves as the coordination mechanism of our competitive environment, dynamically tracking the relative strength of all agents and guiding the co-evolutionary process.

**Elo Rating Updates.** Each agent maintains an Elo rating $R(\cdot)$ that evolves through competition. After a batch of competitions where policy $\pi$ faces various opponents with outcomes $\{S_i\}$, the

policy's rating is updated as:

$$R_{t+1}(\pi) = R_t(\pi) + \sum_{i=1}^{N} K \cdot (S_i - E_{\pi, M_i}) \tag{3}$$

where $K$ is the *K-factor* controlling how strongly each match outcome shifts the rating. A larger $K$ makes the rating respond quickly to wins/losses but also increases variance. $S_i \in \{0, 1\}$ denotes the match outcome (1 if the policy wins, 0 otherwise). $E_{\pi, M_i} = \left(1 + 10^{(R(M_i) - R(\pi))/400}\right)^{-1}$ is the expected win-rate reflecting the theoretical strength gap between the policy and opponent $M_i$.

**Adaptive Curriculum Learning Through Temperature-Controlled Sampling.** The opponent selection follows a temperature-controlled softmax distribution:

$$p(M_k|\pi) \propto \exp\left(-\frac{|R(\pi) - R(M_k)|}{T}\right) \tag{4}$$

where $T$ is a *temperature coefficient* controlling diversity in opponent selection:

- Small $T$ yields a sharp distribution, focusing training on opponents whose Elo ratings are closest to the policy, thus providing a narrow, curriculum-like progression.
- Large $T$ flattens the distribution, increasing opponent diversity but reducing the focus on near-optimal challenge.

This implements an automatic curriculum-learning mechanism: early in training the policy mainly faces similar-strength opponents, and as its Elo rating increases, it automatically transitions to stronger opponents, ensuring training difficulty remains in the optimal challenge regime.

## 2.3 BINARY COMPETITIVE REWARDS WITH GRPO

We adopt the GRPO objective, which removes the value function/critic and estimates advantages from *group-normalized rewards*. For each question $q$, we sample a group of $G$ outputs $\{o_i\}_{i=1}^{G}$ from the old policy $\pi_{\text{old}}$. Each output receives a scalar reward $r_i$ (defined below). GRPO maximizes a PPO-style (Schulman et al., 2017) clipped objective with a KL regularizer to a reference policy $\pi_{\text{ref}}$:

$$J_{\text{GRPO}}(\theta) = \mathbb{E}_{q, \{o_i\} \sim \pi_{\text{old}}} \left[ \frac{1}{G} \sum_{i=1}^{G} \min\left( \frac{\pi_\theta(o_i \,|\, q)}{\pi_{\text{old}}(o_i \,|\, q)} A_i, \text{ clip}\left(\frac{\pi_\theta(o_i \,|\, q)}{\pi_{\text{old}}(o_i \,|\, q)}, 1 - \epsilon, 1 + \epsilon\right) A_i \right) - \beta \, D_{\text{KL}}\left(\pi_\theta \,\|\, \pi_{\text{ref}}\right) \right] \tag{5}$$

where $\epsilon$ controls the clipping range to prevent large policy updates, and $\beta$ regulates the KL divergence penalty to maintain proximity to the reference policy.

**Binary Competitive Rewards.** In our competitive setting, we define the per-output reward $r_i$ using an LLM judge that compares $o_i$ against an opponent's response on the same prompt:

$$r_i = \mathbf{1}\{ J(q, o_i, o^{(\text{opp})}) = \text{policy wins} \} \in \{0, 1\}. \tag{6}$$

These binary rewards are then group-normalized within each batch to compute advantages:

$$A_i = \frac{r_i - \text{mean}(\{r_j\}_{j=1}^{G})}{\text{std}(\{r_j\}_{j=1}^{G})} \tag{7}$$

## 3 THEORETICAL ANALYSIS

Our framework, **Elo-Evolve**, is motivated by two fundamental theoretical advantages of relative comparison over absolute scoring.

**Superior Sample Complexity.** Foundational results in PAC (Probably Approximately Correct) learning theory establish that learning from pairwise comparisons is significantly more sample-efficient than learning from absolute scores (Valiant, 1984; Vapnik & Chervonenkis, 1971; Anthony

---

**Algorithm 1** Elo-Evolve Framework

---

**Require:** Base policy $\pi_0$, Opponent pool $\mathcal{M} = \{M_1, ..., M_K\}$, RM model $J$, Prompts $\mathcal{D}$, Temperature $T$

1: Initialize Elo ratings: $R(\pi_0) = 1350$, $R(M_k)$ based on initial capability estimates
2: **for** each training iteration $t = 0, 1, ...$ **do**
3:      Sample batch of prompts $\{q_i\}_{i=1}^B$ from $\mathcal{D}$
4:      **for** each prompt $q_i$ in batch **do**
5:          Generate policy outputs: $\{o_{i,j}\}_{j=1}^G \sim \pi_t(\cdot|q_i)$
6:          Select opponent via temperature-controlled sampling: $M_i \sim p(M|\pi_t)$ using Eq. (4)
7:          Retrieve opponent response: $o_{M,i}$ from precomputed cache
8:          **for** each policy output $o_{i,j}$ **do**
9:             Evaluate pairwise comparison: $r_{i,j} = \mathbf{1}\{J(q_i, o_{i,j}, o_{M,i}) = \text{policy wins}\}$
10:          **end for**
11:          Compute group-normalized advantages: $A_{i,j} = \frac{r_{i,j} - \bar{r}_i}{\sigma_{r_i}}$
12:      **end for**
13:      Update policy via GRPO objective (Eq. 5) using advantages $\{A_{i,j}\}$
14:      Update Elo ratings: $R_{t+1}(\pi) \leftarrow R_t(\pi) + K \cdot \sum_i (S_i - E_{\pi, M_i})$
15: **end for**

---

& Bartlett, 1999). To achieve a desired ranking error tolerance of $\epsilon$, pairwise learning requires samples on the order of $O(1/\epsilon)$, whereas learning to regress absolute scores to the same precision requires samples on the order of $O(1/\epsilon^2)$ . This quadratic improvement is especially critical in the large-scale, high-precision regime of LLM alignment, enabling faster convergence and more efficient use of high-quality comparison data.

**Inherent Noise Resilience.** Beyond sample efficiency, direct comparison offers superior resilience to the noise inherent in reward signals. We can model the noise characteristics of both paradigms under an idealized, unbiased assumption:

- An **absolute reward model** provides a noisy score $r(y) = q(y) + \epsilon_{\text{abs}}$, where $q(y)$ is the true quality and the scoring noise $\epsilon_{\text{abs}} \sim \mathcal{N}(0, \sigma_{\text{abs}}^2)$. Ranking derived from two such independent scores, $r(y_A)$ and $r(y_B)$, is subject to an effective comparison noise with variance $2\sigma_{\text{abs}}^2$.

- A **direct comparison model** makes a probabilistic judgment $P(y_A \succ y_B) = \Phi\left(\frac{q(y_A) - q(y_B)}{\sigma_{\text{comp}}}\right)$, where $\Phi(\cdot)$ is the standard normal CDF and $\sigma_{\text{comp}}$ is the intrinsic comparison noise.

Under these models, the ranking error rate of the absolute method is determined by the signal-to-noise ratio $\frac{\Delta q}{\sqrt{2}\sigma_{\text{abs}}}$, while the relative method is determined by $\frac{\Delta q}{\sigma_{\text{comp}}}$. Consequently, direct comparison yields a lower ranking error and is considered superior if its intrinsic noise is less than the effective noise of the indirect method. This yields the superiority condition:

$$\sigma_{\text{comp}} < \sqrt{2}\sigma_{\text{abs}} \tag{8}$$

This inequality provides a direct, empirically verifiable criterion for the superiority of relative learning. In Section 5.2, we will experimentally measure both the intrinsic comparison noise ($\sigma_{\text{comp}}$) and the absolute scoring noise ($\sigma_{\text{abs}}$) for a 14B RM, providing strong evidence that our direct comparison approach offers a higher-fidelity training signal in practice.

## 4 ALGORITHM

Our practical implementation addresses the computational challenges of dynamic multi-agent training through several key design choices:

**Pre-computed Response Cache:** To mitigate computational overhead from concurrent opponent model inference, we pre-generate and cache responses from all opponents across the training prompt

set. This transforms expensive model queries into fast dictionary lookups while maintaining the diversity of opponent responses.

**Per-Sample Opponent Selection:** Rather than using a single opponent for the entire batch, we implement per-sample opponent selection within each batch. This enables fine-grained curriculum adaptation and smoother opponent transitions, as different prompts are paired with different opponents based on the current Elo-based sampling distribution, improving both learning efficiency and training stability.

# 5 EXPERIMENTS

## 5.1 EXPERIMENTAL SETUP

### 5.1.1 MODELS AND COMPETITIVE ENVIRONMENT SETUP

**Policy Model:** We use Qwen2.5-7B-Instruct (Hui et al., 2025) as our base policy model $\pi_0$, providing a strong foundation for alignment training while maintaining computational efficiency.

**Opponent Pool:** We construct a diverse competitive environment with opponents of varying capabilities: Qwen2.5-14B-Instruct (initial Elo: 1400), Qwen2.5-32B-Instruct (initial Elo: 1700), and Qwen3-8B-Instruct (Yang et al., 2025) (initial Elo: 2000). Initial Elo ratings are assigned based on model size and capability estimates, providing appropriate starting points for the adaptive rating system.

**RM Model:** All pairwise comparisons are evaluated using Qwen3-14B-Instruct with carefully designed prompts to ensure reliable and consistent win/loss decisions across different response types.

### 5.1.2 DATASET

We train on the Ultra-Feedback dataset (Cui et al., 2023), which contains diverse prompts spanning instruction-following, reasoning, and creative writing tasks, providing comprehensive coverage for evaluating alignment capabilities. Model performance is assessed using Alpaca Eval 2.0 (Dubois et al., 2023), which measures instruction-following quality and response helpfulness through both win-rate and length-controlled metrics, and MT-Bench (Zheng et al., 2023), which evaluates multi-turn dialogue and complex reasoning capabilities across diverse conversational scenarios.

### 5.1.3 TRAINING STRATEGIES COMPARED

We design our experiments to provide progressive validation of our approach through four training paradigms. Point-based Training uses the traditional BT model to convert human preferences into absolute scalar scores, with policy optimization via GRPO maximizing these scores (WorldPM (Binghai Wang & Lin, 2025) as the RM in our implementation). DNO (Rosset et al., 2024) compares self-generated responses against a fixed strong opponent (Qwen2.5-14B in our implementation), using winning responses as positive examples and losing responses as negative examples, optimized with contrastive loss. Note that we implemented DNO ourselves as the original work did not evaluate on Qwen2.5-7B. Static Pairwise Training employs competitive learning against a single fixed opponent throughout the entire training process, using binary win/loss rewards from pairwise comparisons optimized via GRPO. Elo-Evolve represents our proposed framework with Elo-orchestrated adaptive opponent selection, dynamically adjusting training difficulty through temperature-controlled sampling. This experimental design isolates the contributions of pairwise comparison over absolute scoring, optimization methodology within pairwise frameworks, and dynamic opponent selection over fixed opponents.

### 5.1.4 IMPLEMENTATION DETAILS

**Training Configuration:** We implement our approach using the VerL framework with GRPO optimization. Key hyperparameters include: batch size 128, learning rate $1 \times 10^{-6}$, maximum sequence length 4096, KL coefficient $\beta = 0.001$, and Elo K-factor 32.

Table 1: Performance comparison at different training steps. Results show Alpaca Eval 2.0 (WR/LC) and MT-Bench scores. All Qwen models are Instruct versions. Numbers in italics indicate training steps. **Bold**: best results; underlined: second-best results.

| Method | Alpaca Eval 2.0 (WR/LC) | | | MT-Bench | | |
|---|---|---|---|---|---|---|
| Qwen2.5-7B (base model) | 33.35/33.59 | | | 7.84 | | |
| | *100* | *300* | *500* | *100* | *300* | *500* |
| Point GRPO | 41.30/34.95 | 47.76/33.23 | 49.01/37.41 | 7.81 | 7.91 | 7.79 |
| DNO (replicated) | 32.55/31.74 | 33.23/33.18 | 32.48/32.20 | 7.95 | 7.92 | 7.97 |
| vs. Qwen2.5-14B | **46.40**/35.11 | 45.84/34.98 | 48.20/35.84 | 7.98 | 7.99 | **7.99** |
| vs. Qwen2.5-32B | 45.90/**36.18** | 47.20/34.46 | **51.18**/35.55 | 7.79 | 7.96 | 7.89 |
| vs. Qwen3-8B | 44.04/35.90 | 44.22/32.63 | 46.46/34.26 | 7.81 | **8.15** | 7.86 |
| **Elo-Evolve** | 46.21/36.07 | **48.07/35.02** | **51.18/38.03** | **8.03** | 8.04 | 7.82 |

**Computational Optimizations:** To ensure training efficiency, we pre-compute and cache all opponent responses, transforming expensive model inference into fast dictionary lookups. We implement distributed training across 8 GPUs with tensor parallelism.

**Length Bias Mitigation:** To ensure fair evaluation and prevent length-based gaming strategies, we implement a length constraint mechanism: when the policy's response exceeds the opponent's response by more than 300 words, the reward is automatically set to 0. This prevents the policy from exploiting potential judge biases toward longer responses and ensures that improvements reflect genuine quality gains rather than superficial length inflation.

### 5.1.5 MAIN RESULTS: PROGRESSIVE PERFORMANCE GAINS

Tables 1 demonstrate clear performance improvements across our three-tier experimental design, providing strong empirical validation for both the theoretical benefits of pairwise comparison and the practical advantages of dynamic opponent selection.

**Point-based vs Pairwise Baselines:** Point GRPO represents traditional BT absolute scoring, showing moderate but unstable performance (49.01/37.41 at Step 500 declining from 47.76/33.23 at Step 300). DNO shows consistently lower performance (32.48/32.20 at Step 500), illustrating that static opponent selection limits learning potential even within the pairwise paradigm.

**Static Pairwise Training:** Single-opponent configurations demonstrate clear advantages over point-based methods but with significant variability. vs. Qwen2.5-14B maintains stable performance (46.40→48.20 WR across steps), while vs. Qwen2.5-32B shows strong peak performance (51.18 WR at Step 500). However, individual static opponents cannot consistently excel across all metrics and training phases.

**Elo-Evolve:** Our dynamic multi-opponent approach achieves the best or second-best performance in most categories. Notably, Elo-Evolve reaches 51.18/38.03 (WR/LC) at Step 500, matching the best static configuration while maintaining superior consistency. In MT-Bench, our method leads at Steps 100 (8.03) and 300 (8.04) but shows a decline at Step 500 (7.82). This decline reflects our adaptive opponent selection mechanism: at Step 500, our primary opponent became Qwen3-8B, which itself degraded significantly (8.15→7.86), causing our model to adapt to this weakened opponent. This demonstrates both the responsiveness of our system and suggests potential improvements in opponent pool management. Excluding the Step 500 MT-Bench anomaly explained above, Elo-Evolve maintains remarkable consistency and leadership across different training phases and evaluation metrics. This stability, combined with peak performance achievements, validates the effectiveness of dynamic opponent selection over fixed strategies.

### 5.1.6 SCALABILITY AND GENERALIZATION ANALYSIS

Table 2 demonstrates the broad applicability of competitive learning across varying opponent capabilities and model families, establishing our framework's versatility for diverse scenarios.

Table 2: Scalability analysis: Pairwise training against diverse opponents. Results demonstrate consistent improvements across different opponent capabilities and model families.

| Opponent Configuration | Alpaca Eval 2.0 (WR/LC) | | | MT-Bench | | |
|---|---|---|---|---|---|---|
| Qwen2.5-7B (base model) | 33.35/33.59 | | | 7.84 | | |
| | *100* | *300* | *500* | *100* | *300* | *500* |
| *Training against weaker opponents* | | | | | | |
| vs. Qwen2.5-1.5B | 38.45/**37.18** | 39.75/**37.52** | 37.64/**35.72** | **7.98** | **8.13** | 7.76 |
| *Training against same-capacity opponents* | | | | | | |
| vs. Qwen2.5-7B | **44.47**/33.29 | **47.83**/33.92 | **49.19**/33.07 | 7.94 | 8.09 | **8.05** |
| *Training against different model families* | | | | | | |
| vs. Llama-3.1-70B | 43.54/36.22 | 46.58/35.41 | 47.83/31.86 | 7.89 | 8.02 | 7.90 |

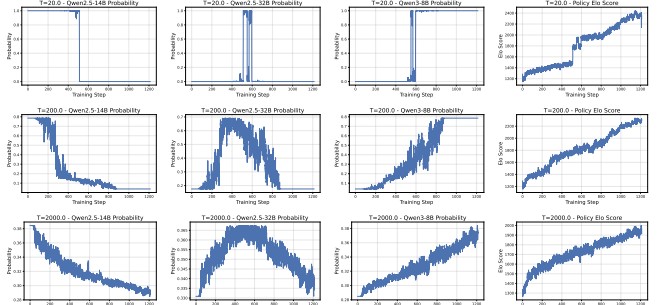
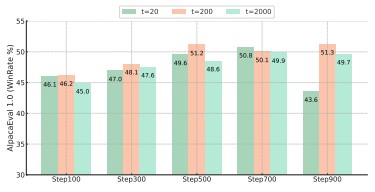

Figure 1: AlpacaEval WR performance for different $T$ values.

Figure 2: Comparison of opponent sampling probabilities and Elo rating evolution for three temperature settings ($T = 20$, $T = 200$, $T = 2000$). Each row shows a different temperature setting; columns show 14B, 32B, 8B opponent probabilities and policy Elo.

**Weaker Opponents:** Competition with Qwen2.5-1.5B consistently improves over baseline performance (38.45/37.18 vs base 33.35/33.59), proving that even substantially weaker opponents provide valuable learning signals. This occurs through encouraging clearer articulation, more confident responses, and basic competency validation—essential for robust policy development.

**Same-Capacity Opponents:** Training against Qwen2.5-7B yields exceptionally strong WinRate performance (44.47→49.19 across steps), suggesting same-capacity competition drives nuanced policy refinements. Equal-strength opponents expose subtle weaknesses and encourage sophisticated improvements that larger capability gaps might mask.

**Cross-Family Opponents:** Results with Llama-3.1-70B validate our framework's architecture-agnostic nature. Despite the 10× parameter disadvantage and different training methodologies, competitive learning produces substantial improvements (43.54/36.22 vs baseline), confirming cross-family applicability and robustness to architectural differences.

Each opponent configuration demonstrates distinct advantages: weaker opponents (1.5B) provide foundational improvements, same-capacity opponents (7B) excel in nuanced refinements, and cross-family opponents (Llama-70B) validate architectural generalization. This diversity of benefits suggests that different opponent types contribute complementary learning signals, supporting our multi-opponent competitive framework.

### 5.1.7 TEMPERATURE PARAMETER ANALYSIS

Figure 2 provides comprehensive insights into how temperature parameter T affects both opponent selection dynamics and learning outcomes. The left panel shows opponent sampling probabilities and Elo evolution across training. Figure 1 demonstrates the corresponding performance trajectories.

**Greedy Selection (T=20):** The low temperature creates sharp, deterministic opponent transitions. At Step 500, the selection abruptly switches from 14B (probability 1→0) to 32B (0→1→0 by Step

600), then exclusively focuses on Qwen3-8B. While this achieves the highest final Elo rating (2400), it leads to catastrophic performance degradation at Step 900 (50.8→43.6), when the dominant opponent (Qwen3-8B) itself deteriorates. This demonstrates the brittleness of overly focused opponent selection.

**Optimal Balance at T=200:** The moderate temperature enables smooth, gradual transitions between opponents. The 14B probability slowly decreases (0.78→0.03), 32B rises then falls (0.1→0.7→0.1), and Qwen3-8B gradually increases (0.03→0.78). This balanced progression achieves strong performance throughout training and reaches a competitive final Elo (2300), validating our temperature-controlled sampling mechanism. The smooth transitions prevent over-dependence on any single opponent while maintaining curriculum learning benefits.

**Random Selection (T=2000):** High temperature maintains the same opponent transition trends as T=200 (14B decreasing, 32B rising then falling, Qwen3-8B increasing) but with severely dampened amplitude—all probabilities are constrained to oscillate within 0.3-0.4 range. This flattened selection distribution prevents the system from sufficiently focusing on appropriate-difficulty opponents during critical learning phases, resulting in the lowest final Elo (2000) and consistently suboptimal performance. While the overall curriculum progression is preserved, the reduced selection intensity fails to provide adequate learning signals.

The temperature parameter critically balances curriculum learning focus with opponent diversity. T=20 maximizes Elo progression but creates fragility; T=2000 provides stability but sacrifices learning efficiency; T=200 achieves the optimal trade-off between adaptive curriculum and robust performance. The smooth opponent transitions at T=200 demonstrate how proper calibration enables natural learning progression without catastrophic failures when individual opponents degrade.These results confirm that effective adaptive opponent selection requires careful temperature calibration to achieve both strong learning dynamics and robust performance maintenance.

### 5.1.8    FUTURE APPLICATIONS TO VERIFIABLE TASKS

Our competitive framework shows particular promise for tasks with verifiable rewards, such as mathematical reasoning, code generation, and formal verification (RLVR scenarios). Traditional GRPO training faces a fundamental limitation: when all sampled responses are uniformly correct or incorrect, gradient signals vanish, wasting valuable training data.

Our competitive approach addresses this limitation elegantly. Even when responses achieve identical correctness, pairwise comparison can evaluate nuanced quality dimensions—reasoning clarity, solution elegance, computational efficiency, or explanation completeness. For instance, in mathematical problem-solving, two correct solutions can be differentiated by proof conciseness, pedagogical clarity, or methodological sophistication.

This capability transforms binary correctness into rich, multi-dimensional feedback, maximizing data utilization and enabling continuous improvement even in high-accuracy regimes. Future work should explore this application to complex reasoning domains where solution quality extends far beyond mere correctness.

### 5.2    NOISE ANALYSIS IN REWARD SIGNALS

To ground our theoretical claims, we empirically analyzed the noise levels in both absolute and relative reward signals using a rigorously constructed dataset of creative writing responses. Our dataset consists of responses with expert-annotated quality scores, where three domain experts participated in the annotation process: two experts performed initial independent annotations on a 1-5 quality scale, and a third expert conducted secondary validation. The inter-annotator agreement reached 81.5%, indicating high annotation reliability for this inherently subjective creative writing task.

Using this expert-validated dataset, we prompted Qwen3-14B-Instruct to perform two evaluation tasks: (1) absolute scoring of individual responses and (2) direct pairwise comparison of response pairs across different quality gaps.

We estimated the noise parameters for both approaches:

- **Effective Absolute Ranking Noise ($\sigma_{abs,eff}$):** This metric represents the equivalent noise level when using absolute scores to perform ranking. It accounts for both signal compression and random noise in the LLM's scoring behavior.

- **Intrinsic Comparison Noise ($\sigma_{comp}$):** We estimated this parameter using maximum likelihood estimation under the Thurstone model for different quality gaps.

Our analysis reveals a substantial difference in noise levels:

- Effective Absolute Ranking Noise: $\sigma_{abs,eff} \approx \mathbf{35.65}$

- Intrinsic Comparison Noise (Gap 1): $\sigma_{comp} \approx \mathbf{7.85}$

The effective noise of ranking via absolute scores is **4.5 times higher** than that of direct pairwise comparison. This finding provides strong empirical evidence that direct comparison offers a significantly higher-fidelity training signal, especially crucial for subjective tasks like creative writing where quality assessment is inherently challenging.

## 6  RELATED WORK

The landscape of LLM alignment has rapidly evolved beyond traditional RLHF (Christiano et al., 2017; Ouyang et al., 2022). Early approaches distilled human preferences into scalar reward models then optimized policies via reinforcement learning, showing promise but suffering from reward hacking and instability. Subsequent work like Constitutional AI (Bai et al., 2022) and RLAIF (Lee et al.) scaled supervision through AI judges but maintained absolute scoring paradigms.

Direct Preference Optimization (DPO) (Rafailov et al., 2023) eliminates explicit reward models by directly optimizing policy preferences through a contrastive objective, avoiding the complexities of two-stage RLHF. Later extensions explored listwise variants and improved sampling schemes (Liu et al., 2024). Direct Nash Optimization (DNO) (Rosset et al., 2024) frames preference learning as finding Nash equilibria in two-player zero-sum games, using self-play with large-margin win-loss pairs and regression-based objectives rather than reinforcement learning. While DNO supports off-policy data and provides theoretical convergence guarantees, it relies on a fixed preference oracle (*e.g.*, GPT-4), limiting adaptability as the student policy improves. Relative Preference Optimization (RPO) (Yin et al., 2024) extends preference learning beyond single-prompt constraints by leveraging semantic similarity to enable cross-prompt comparisons. RPO constructs offline contrast matrices weighted by semantic similarity, but lacks dynamic difficulty adjustment mechanisms as the policy evolves. Recent self-play approaches (Whitehouse et al., 2025; Wang et al., 2025) eliminate external supervision entirely by generating win/loss labels from the learner's own outputs. While appealingly simple, these methods suffer from a fundamental ceiling effect: once the policy surpasses its own best responses, the training distribution collapses and further progress stalls due to the absence of stronger external anchors.

Elo-Evolve distinguishes itself by treating alignment as a multi-agent competition with adaptive opponent management. Our framework maintains a diverse opponent pool with Elo-based sampling that automatically adjusts difficulty while preserving strong external anchors, avoiding both the rigidity of static approaches and the ceiling effects of pure self-play methods. Unlike fixed oracle approaches, our dynamic mechanism ensures continuous challenge adaptation throughout training.

## 7  CONCLUSION

We introduced Elo-Evolve, a co-evolutionary framework that fundamentally redefines LLM alignment from static reward optimization to dynamic multi-agent competition. Our theoretical analysis demonstrates that pairwise comparison enjoys superior sample complexity compared to absolute scoring approaches, with empirical validation showing a $4.5\times$ noise reduction. Experimental results validate our progressive experimental design, demonstrating clear performance improvements from absolute scoring through static pairwise training to our dynamic opponent selection approach.

Our framework demonstrates the potential of competitive learning approaches for LLM alignment, opening directions for future work in multi-agent training and adaptive curriculum design.

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

## A  USE OF LARGE LANGUAGE MODELS

In accordance with the conference guidelines, we disclose that Large Language Models were used to assist in the preparation of this manuscript. Specifically, we employed LLMs for writing assistance, including improving grammar, clarity, and academic writing style throughout the paper.

We emphasize that all LLM suggestions were carefully reviewed and manually edited by the authors. LLM-generated content was not used verbatim; rather, LLM feedback served as input for human-authored revisions. The authors take full responsibility for all technical content, experimental design, results, and conclusions presented in this work. All core contributions, theoretical analysis, experimental methodology, and scientific claims are the original work of the authors.

## B  EMPIRICAL NOISE ANALYSIS: DETAILED METHODOLOGY AND RESULTS

### B.1  EXPERIMENTAL SETUP

To empirically validate our theoretical claims about the superior noise characteristics of pairwise comparison over absolute scoring, we conducted a comprehensive noise analysis using expert-annotated creative writing data and LLM evaluations.

**Dataset Construction:** Our evaluation focuses on creative writing responses, which represent one of the most challenging domains for quality assessment due to their subjective nature. We constructed a dataset of 1,086 creative writing responses spanning various genres and complexity levels. The quality annotation process involved three domain experts:

- *Primary Annotation:* Two experts independently annotated each response on a 1-5 quality scale, considering criteria such as creativity, coherence, language quality, and thematic depth.

- *Secondary Validation:* A third expert reviewed all annotations, resolving discrepancies and ensuring consistency.

- *Quality Control:* The inter-annotator agreement reached 81.5%, demonstrating high reliability despite the inherent subjectivity of creative writing evaluation.

From this annotated dataset, we derived two evaluation datasets: (1) 1,086 responses for absolute scoring analysis, and (2) 1,037 response pairs for pairwise comparison analysis across different quality gaps ($\Delta q \in \{1, 2, 3, 4\}$).

All evaluations were performed by Qwen3-14B-Instruct using carefully designed prompts based on expert annotation criteria. Each response received 5 independent absolute ratings and 5 independent comparison judgments to ensure statistical reliability.

## B.2 ABSOLUTE SCORING ANALYSIS

We analyzed the absolute scoring behavior using a comprehensive statistical model that accounts for both random noise and systematic biases.

Using linear regression between expert quality scores and LLM ratings, we found:

- Signal compression factor (slope): $a = 0.028$
- Bias offset (intercept): $b = 2.85$
- R-squared: $0.003$

The extremely low slope indicates severe signal compression—the LLM's effective scoring range is dramatically narrower than the true quality variation. The near-zero R² reveals that LLM scores have virtually no correlation with expert-annotated quality.

The LLM's scoring distribution shows severe bias:

- Score 1: 16.1% (880/5,430 ratings)
- Score 2: 15.1% (822/5,430 ratings)
- Score 3: 41.9% (2,275/5,430 ratings)
- Score 4: 14.0% (760/5,430 ratings)
- Score 5: 12.8% (693/5,430 ratings)

The concentration in middle scores (particularly score 3) demonstrates the model's reluctance to make discriminative judgments. Noise Estimation:

- Within-sample variance: $\sigma_{\text{within}}^2 = 0.942$
- Residual standard deviation: $\sigma_{\text{residual}} = 0.707$
- Effective ranking noise: $\sigma_{\text{abs,eff}} = \frac{\sqrt{2} \cdot \sigma_{\text{residual}}}{a} = 35.65$

## B.3 PAIRWISE COMPARISON ANALYSIS

We estimated comparison noise using maximum likelihood estimation under the Thurstone model, analyzing performance across different quality gaps. Gap-Stratified Results:

- Gap 1 ($\Delta q = 1$): $\sigma_{\text{comp}} = 7.85$, accuracy = 55.1% (288 pairs, 864 comparisons)
- Gap 2 ($\Delta q = 2$): $\sigma_{\text{comp}} = 5.80$, accuracy = 63.5% (209 pairs, 627 comparisons)
- Gap 3 ($\Delta q = 3$): $\sigma_{\text{comp}} = 8.13$, accuracy = 64.4% (125 pairs, 375 comparisons)
- Gap 4 ($\Delta q = 4$): $\sigma_{\text{comp}} = 25.53$, accuracy = 56.2% (53 pairs, 159 comparisons)

Even for the most challenging scenario (Gap 1, minimal quality difference), pairwise comparison achieves 55.1% accuracy—above random chance and with substantially lower noise ($\sigma_{\text{comp}} = 7.85$) compared to absolute scoring. The performance peaks at Gap 2, suggesting an optimal discrimination range for the model.

### B.4 COMPARATIVE ANALYSIS AND IMPLICATIONS

Comparing the most challenging pairwise scenario (Gap 1) with the effective absolute ranking noise:

$$\frac{\sigma_{\text{abs,eff}}}{\sigma_{\text{comp}}} = \frac{35.65}{7.85} = 4.54$$

This **4.5× noise reduction** provides strong empirical support for our theoretical framework, demonstrating the substantial advantage of pairwise comparison in noisy evaluation scenarios.

The absolute scoring method exhibits two critical failures:

1. *Signal Compression:* With $a = 0.028$, the effective scoring range is compressed by 97%, discarding most quality information.
2. *Discriminative Failure:* The near-zero correlation ($R^2 = 0.003$) with expert judgments indicates the scores contain essentially no quality signal.

In contrast, pairwise comparison maintains discriminative power even in the most challenging conditions, with accuracy consistently above random chance across all quality gaps.

