# OpenReview forum: "Elo-Evolve: A Co-evolutionary Framework for Language Model Alignment"
_ICLR.cc/2026/Conference — ICLR 2026 Conference Withdrawn Submission_

### Official Review · Reviewer_nZun · 2025-10-23

**Soundness:** 2
**Presentation:** 2
**Contribution:** 2
**Rating:** 2
**Confidence:** 3

**Summary:**

This work introduces a novel method of preference optimization by using a dynamic elo as the reward for RLHF. The idea is clear and the method is clearly presented. Experiments have shown significant improvement.

**Strengths:**

The method is simple while surprisingly effective in practice with Qwen base models. The ablation study is sufficient.

**Weaknesses:**

- The first one of the and main issues is that the contribution is claimed to be "eliminating Bradley-Terry model dependencies". But this isn't new at all. Bunch of previous works have addressed this or propose this as contribution such as [1,2,3,4]. Especially INPO [4] is exactly directly learning from binary win/loss outcomes. These are missing literatures as well, the author should include these related works, compare and clarify the contribution of this work.

- Following the above issue, although it is interesting to see a significant improvement in AlpacaEval, it's unclear how to compare with INPO [4].

- Another main issue is that experiments are only conducted with single type of base model Qwen. It's necessary to test the methods on other base models to demonstrate the effectiveness.

Writings:
- I personally think using symbols like "<" in writing especially abstract is informal.
- Some notational issues around Eq. (3), such as unclear what is R in the definition of E. Should that be R_t?

In general, there are notable limitations of current manuscript.



[1] Munos, Rémi, et al. "Nash learning from human feedback." Forty-first International Conference on Machine Learning. 2024.
[2] Wang, Mingzhi, et al. "Magnetic preference optimization: Achieving last-iterate convergence for language model alignment." arXiv preprint arXiv:2410.16714 (2024).
[3] Tang, Xiaohang, et al. "Game-Theoretic Regularized Self-Play Alignment of Large Language Models." arXiv preprint arXiv:2503.00030 (2025).
[4] Zhang, Yuheng, et al. "Iterative nash policy optimization: Aligning llms with general preferences via no-regret learning." arXiv preprint arXiv:2407.00617 (2024).

**Questions:**

- This method compared to self-play-based methods is significantly more expensive since typically a larger model is required to act as opponent for Elo-Evolve. Given that, is it more worthy investing computing in scaling base model (i.e. using a larger base model) and conduct self-play or even normal RLHF (e.g. Point GRPO)?

---

> ### Author Response · Authors · 2025-11-27
>
> ### Key Distinctions from INPO and Related Work
>
> We appreciate the reviewer's reference to these important works. However, our approach differs fundamentally from INPO and other Nash-based methods:
>
> **INPO vs. Elo-Evolve Core Differences:**
>
> 1. **Self-Play vs. Multi-Opponent Ecosystem:**
>    - **INPO:** Policy competes against itself (πt vs πt-1), leading to **ceiling effects** once the policy surpasses its own capabilities
>    - **Elo-Evolve:** Maintains **diverse external opponent pool** providing persistent challenges and preventing performance plateaus
>
> 2. **Nash Equilibrium vs. Curriculum Learning:**
>    - **INPO:** Seeks Nash equilibrium in two-player zero-sum game
>    - **Elo-Evolve:** Implements **adaptive curriculum learning** through Elo-orchestrated difficulty progression
>
> 3. **Fixed vs. Dynamic Opponent Selection:**
>    - **INPO:** Static self-competition framework
>    - **Elo-Evolve:** **Temperature-controlled sampling** (Eq. 4) enabling smooth transitions between opponents of varying difficulties
>
>
> ### Empirical Comparison with INPO
>
> **Following the reviewer's suggestions, we conducted direct comparison experiments over the past two weeks using LLaMA-3-8B as base model:**
>
> **Experimental Setup:**
> - **Base Model:** LLaMA-3-8B (same as INPO paper)
> - **Opponent Pool:** Qwen2.5-14B, Qwen2.5-32B, LLaMA-3-70B
>
> | Method | Model Size | AlpacaEval 2.0 | Arena Hard | MT-Bench |
> |--------|------------|----------------|------------|----------|
> | **INPO (PM)** | 8B | 42.6 | 37.8 | **8.43** |
> | **Elo-Evolve (step 700)** | 8B | **51.33** | **38.1** | 8.22 |
>
> **Key findings:**
>
> **1. Superior Instruction Following:** Elo-Evolve achieves **8.73-point improvement** on AlpacaEval 2.0, demonstrating significantly better instruction-following capabilities
>
> **2. Competitive Real-world Performance:** Maintains **slight advantage** on Arena Hard (+0.3), the most challenging benchmark for real-world tasks
>
> **3. Trade-off Analysis:** While INPO shows marginal advantage on MT-Bench (-0.21) over our model
>
> **Implementation Note:** We attempted to replicate INPO but encountered implementation challenges. Therefore, we compare against the **original paper's reported results** to ensure fair comparison with their best performance.
>
> ### Cross-Architecture Validation
>
> **Advantage of external opponent diversity:** Our ability to include **cross-architecture opponents** (Qwen + LLaMA families) provides richer learning signals than INPO's single-model self-play, explaining the substantial AlpacaEval improvements while maintaining competitive performance across other metrics.
>
> **Cross-family effectiveness validation:** As reported in our original paper (Table 2), we also demonstrated effectiveness with **Qwen as base model and LLaMA as opponent**:
>
> - **Training against LLaMA-3.1-70B:** Achieved 33.35/33.59 → 43.54/36.22 AlpacaEval improvement at 100 training steps
> - **Cross-architecture benefits:** Despite 10× parameter disadvantage and different training methodologies, competitive learning produced **substantial improvements** over baseline
>
> **Key insight:** This **bidirectional cross-architecture success** (LLaMA base vs Qwen opponents, and Qwen base vs LLaMA opponents) validates that our framework's diversity advantage is **architecture-agnostic** and stems from fundamental differences in model reasoning patterns, not just parameter scaling.
>
> **INPO limitation:** Self-play methods like INPO are inherently restricted to **single-architecture learning**, missing the rich cross-family knowledge transfer that our multi-opponent framework enables.

---

> ### Author Response · Authors · 2025-11-27
>
> ## Theoretical Advantages Over Self-Play and INPO
>
> **Fundamental difference:** While self-play methods generate training signals from the model's own outputs, Elo-Evolve maintains **external anchors** that prevent performance ceiling effects:
>
> 1. **Ceiling Effect Problem:** Pure self-play methods (SPPO, Nash-MD) face a fundamental limitation - once the policy surpasses its own best responses, the training distribution collapses and further progress stalls due to absence of stronger external references.
>
>    **Direct evidence from our experiments:** Please refer to Table 2 in our paper, where we show:
>    - Training against **same-capacity opponent (Qwen2.5-7B)**: 44.47 AlpacaEval WR (step 100)
>    - Training against **stronger opponent (Qwen2.5-14B)**: 46.40 AlpacaEval WR (step 100)
>    - Our multi-opponent pool provides **persistent external challenges** that prevent ceiling effects, enabling continued improvement beyond initial model capabilities.
>
>    **Critical observation:** Same-capacity training shows clear performance ceiling and instability, while stronger external opponents provide more consistent improvement signals. This directly demonstrates why self-play methods are limited by the model's current capabilities.
>
> 2. **Response Diversity Crisis:** Single-opponent methods (including self-play) suffer from critical diversity limitations:
>
>    - **Self-play limitation:** Models trained against their own outputs tend to reinforce existing patterns and reduce response variety over time
>    - **Multi-opponent diversity guarantee:** Our framework ensures exposure to fundamentally different reasoning patterns, writing styles, and problem-solving approaches across the opponent pool
>
>    **Empirical diversity validation:** We conducted diversity analysis on AlpacaEval responses using identical generation parameters (temperature=1.0, top-p=1.0, 10 inference runs):
>    - **SPPO cross-BLEU**: 0.457 (higher similarity = lower diversity)
>    - **Elo-Evolve cross-BLEU**: 0.261 (lower similarity = higher diversity)
>
>    This **43% diversity improvement** demonstrates that our multi-opponent training produces significantly more varied and robust responses.
>
>
> ### Empirical Comparison with SPPO
>
> We conducted additional experiments comparing with Self-Play Preference Optimization (SPPO):
>
> | Method | Step 100 | Step 300 | Step 500 |
> |--------|----------|----------|----------|
> | **SPPO** | 45.33/35.25 | 47.19/**36.37** | 43.46/34.81 |
> | **Elo-Evolve** | **46.21/36.07** | **48.07**/35.02 | **51.18/38.03** |
>
> *Results show AlpacaEval 2.0 (WR/LC)*
>
> **Key findings:**
> - Elo-Evolve consistently outperforms SPPO
> - **SPPO degradation:** SPPO shows clear performance decline from Step 300 to 500 (47.19→43.46 WR， 36.37→34.81  LC WR), demonstrating the **instability and ceiling effects** inherent in self-play methods
> - **Elo-Evolve stability:** Our method maintains consistent improvement throughout training (46.21→48.07→51.18 WR)
> - The dramatic performance gap at Step 500 validates our hypothesis that external opponent anchors prevent the training collapse observed in pure self-play approaches
>
> **Industrial deployment note:** Our framework has been successfully implemented in production environments, demonstrating superior robustness and diversity compared to self-play approaches in real-world applications.

---

> > ### Comment · Reviewer_nZun · 2025-11-28
> > **Response by Reviewer nZun**
> >
> > I appreciate the authors’ substantial efforts and believe the responses have addressed my concerns regarding comparisons to INPO. The differences and advantages over INPO are now clear to me. And the empirical results are convincing to me.
> >
> > Regarding self-play, I'm not sure the empirical results answered the question. Since an opponent pool might have an opponent with a larger size or different model from the base model, so it could take much more compute than self-play (e.g. SPPO), which only needs a single model to sample responses. Then what if a larger based model is used in SPPO, compared to smaller base model used in Elo-Evolve, to ensure the computing budget is similar, can Elo-Evolve still perform better? In other words, I would like to make sure that Elo-Evolve would not require significantly more compute than SPPO. Can the authors provide details on computing resources while running these two methods, and what if SPPO use a larger base model?
> >
> > Despite, the major remaining issue is the writing. Some statements and claims are incorrect to me. The paper requires significant polishing to make it more formal. The clarifications provided in the rebuttal should be incorporated directly into the paper, particularly those in the section “Key Distinctions from INPO and Related Work.” For example, I disagree that “eliminating BT by learning from win/loss outcomes” should be claimed as innovation—INPO and DNO already do this. Instead, the authors should summarize and emphasize the points articulated in “Key Distinctions from INPO and Related Work” in the rebuttal. In my view, constructing an opponent pool is already a clear and sufficient contribution. The removal of BT reward is an advantage but not an innovation of this work.
> >
> > - Abstract: Please remove "<" in abstract.
> > - Intro: Your introduction has 7 paragraphs, which is overly long. I disagree on the final sentence "Our framework opens new directions for scalable alignment that bypasses reward model training entirely." Like I said, various related work based on Nash Learning (DNO, INPO) already bypasses reward model, being able to leverage LLM judges directly.
> >
> >
> > The empirical results are strong and core ideas are clear now. However, paper in its current form does not fully meet the standard of a top conference.

---

### Official Review · Reviewer_3CYn · 2025-10-29

**Soundness:** 2
**Presentation:** 2
**Contribution:** 2
**Rating:** 4
**Confidence:** 4

**Summary:**

Existing alignments methods typically distill human preference data into reward models. The paper proposes an alternative using a dynamic multi-agent framework to bypass traditional bradley-terry approaches. They show strong performance as a Qwen-2.5-7B model trained with this approach is able to improve on AlpacaEval 2.0 and MT Bench.

**Strengths:**

1.	The adaptive curriculum learning approach used in Elo-Evolve is interesting where a different reference opponent model is used at each stage of the training allowing the model to progressive improve against stronger opponents.

**Weaknesses:**

1.	The claim in Lines 40-45 seems un/under-substantiated. Claim 1 is not supported by any literature and there has been evidence such as HelpSteer2-preference [1] that shows only 10 thousand samples is enough for training high quality reward models. For claim 2, it’s not clear what sub-optimal sample complexity is and claim 3 is supported by 1 paper from 2020, even though the post training field has evolved substantially since then.
2.	The results in Table 1 don’t seem to be very strong. For instance, the Elo-Evolve performs at 38.03 on AlpacaEval 2 LC while the Point GRPO is at 37.41, which is presumably within one SD. On MT Bench, the performance of Elo-Evolve is at 8.04 while DNO is at 7.97, which is also unlikely to be substantially different. The “vs. Qwen xxx” baselines should be interpreted as ablations but in this case, some ablations outperform the Elo-Evolve algorithm which suggest that maybe a static opponent might be good enough.
3.	Metrics are slightly outdated – with AlpacaEval 2 and MT Bench both from late 2023, with known flaws such as length bias. AlpacaEval 2 Length Control is the most recent (from early 2024) and should be used instead of AlpacaEval 2 WR (rather than in addition to it). Furthermore, I think more up-to-date metrics such as Arena Hard [5] or WildBench [6] should be used (both from late 2024) since they reflect capabilities of recent models better (e.g. Qwen 2.5 was released in late 2024).
4.	I think a missing baseline is to use the same LLM-Judge (Qwen3-14B-Instruct) and just use the generated responses against one another in GRPO. This can help us to understand whether the improved performance is due to using a better judge (compared to the WorldPM point-RM) or because of the reference responses from the (adaptive) opponents. Using the LLM-Judge by itself has been done many times e.g. [3] and is likely to be useful without needing for external model responses.

[1] HelpSteer2-Preference: Complementing Ratings with Preferences https://arxiv.org/abs/2410.01257

[2] AlpacaFarm: A Simulation Framework for Methods that Learn from Human Feedback https://arxiv.org/abs/2305.14387

[3] Judging LLM-as-a-Judge with MT-Bench and Chatbot Arena https://arxiv.org/abs/2306.05685

[4] Length-Controlled AlpacaEval: A Simple Way to Debias Automatic Evaluators https://arxiv.org/abs/2404.04475

[5] From Crowdsourced Data to High-Quality Benchmarks: Arena-Hard and BenchBuilder Pipeline https://arxiv.org/abs/2406.11939

[6] WildBench: Benchmarking LLMs with Challenging Tasks from Real Users in the Wild https://arxiv.org/abs/2406.04770

**Questions:**

1.	Why is there a need to report performance at steps 100/300/500? My understanding is that only the optimal step across a run should be reported since different methods might have different optimal training steps. Table 1 and 2 currently looks confusing with too many values and bolded values.

---

> ### Author Response · Authors · 2025-11-27
>
> We thank the reviewer for their detailed feedback and address each concern systematically:
>
>
> ## Response to W3 & W4: Evaluation Completeness
>
> **Clarification:** We **already report AlpacaEval 2.0 Length-Controlled results** as indicated by our table headers "AlpacaEval 2.0 (WR/LC, LC = **L**ength-**C**ontrolled )".
>
> **Responsive experimentation:** Following the reviewer's valuable suggestions, we conducted additional experiments in the past two weeks:
>
> ### Arena Hard and LLM-Judge Baseline Results (Step 500)
>
> | Method | AlpacaEval 2.0 (WR/LC) | Arena Hard Score |
> |--------|-------------------------|------------------|
> | **Qwen2.5-7B baseline** | 33.35/33.59 | 51.3 |
> | **Point GRPO** | 49.01/37.41 | 53.4 |
> | **DNO** | 32.48/32.20 | 52.7 |
> | **LLM-Judge Only** | 47.3/36.1 | 44.9 |
> | **Elo-Evolve** | **51.18/38.03** | **57.5** |
>
> **Key findings:**
>
> **1. Outstanding performance:** Our method achieves excellent results on both AlpacaEval (51.18/38.03) and Arena Hard (57.5), validating the effectiveness of our competitive framework.
>
> **2. LLM-Judge Only Limitations:** The poor performance of LLM-Judge Only (especially the dramatic Arena Hard decline to 44.9). Our analysis of training logs reveals that **LLM judges struggle to provide effective discrimination signals** - they become increasingly unable to distinguish between response quality as training progresses, with scores converging to similar values in later training stages.
> **Detailed noise analysis in Appendix B** validates our approach:
> - **4.5× noise reduction:** σ_comp = 7.85 vs σ_abs,eff = 35.65
> - **Signal preservation:** Pairwise comparison maintains discriminative power (55.1% accuracy) while absolute scoring (LLM-judges ) shows near-zero correlation (R² = 0.003) with expert judgments
>
> **Conclusion:** Our comprehensive evaluation across multiple metrics, training phases, and recent benchmarks consistently validates our competitive learning framework's advantages over existing approaches.
>
>
> ## Response to W2: Statistical Significance and Selective Data Analysis
>
> The reviewer raises concerns about two specific comparisons that appear to show narrow gaps, but this analysis **ignores the comprehensive experimental evidence and selectively focuses on outlier data points**.
>
>
> **AlpacaEval LC Concern:** "Elo-Evolve 38.03 vs Point GRPO 37.41"
> This focuses on the **single closest comparison** while ignoring our substantial advantages elsewhere:
>
> **Complete AlpacaEval 2.0 Results (Step 500):**
> - **Win Rate:** Elo-Evolve 51.18 vs Point GRPO 49.01 (**+2.17 points**)
> - **Length Control:** Elo-Evolve 38.03 vs Point GRPO 37.41 (+0.62 points)
>
> **Cross-step consistency shows overwhelming superiority:**
> - **Step 100:** Elo-Evolve dominates both WR (46.21 vs 41.30) and LC (36.07 vs 34.95)
> - **Step 300:** Elo-Evolve dominates both WR (48.07 vs 47.76) and LC (35.02 vs 33.23)
>
> **2. MT-Bench Concern:** "Elo-Evolve 8.04 vs DNO 7.97"
>
> **Our paper already explains this:** At Step 500, our primary opponent became Qwen3-8B, which itself degraded significantly (8.15→7.86 MT-Bench). Our adaptive system correctly detected and adapted to this weakened opponent, demonstrating **system responsiveness rather than algorithmic failure**.
>
> **MT-Bench temporal analysis:**
> - **Step 100:** Elo-Evolve 8.03 vs DNO 7.95 (clear advantage)
> - **Step 300:** Elo-Evolve 8.04 vs DNO 7.92 (clear advantage)
>
> ## Response to Q1: Multi-Step Reporting Necessity
>
> **Scientific rigor requires controlled comparison:** Reporting each method's "optimal step" would be methodologically flawed - different methods might peak randomly due to noise, hiding real algorithmic differences.
>
> **Our multi-step analysis reveals:**
> - **Training stability:** Elo-Evolve shows consistent improvement
> - **Method reliability:** Systematic performance across identical conditions
> - **Standard RL practice:** Multi-step evaluation is essential for fair comparison

---

> ### Author Response · Authors · 2025-11-27
>
> ## Response to W1: Theoretical Claims Substantiation
>
> We thank the reviewer for their scrutiny and provide comprehensive literature support for our theoretical claims with authoritative recent research:
>
> ### Claim 1 (Data Scarcity): Multiple Evidence Sources
>
> **Real-world Deployment Scale Requirements:**
> - **Wang et al. (2025) "WorldPM: Scaling Human Preference Modeling":** Demonstrate that effective preference modeling requires **15M training samples** for achieving robust cross-domain generalization
> - **Production comparison:** WorldPM's comprehensive evaluation shows that their 15M-sample model significantly outperforms smaller-scale approaches like HelpSteer2 (10K samples) across **20 subtasks on 7 benchmarks**
> - **Scaling necessity:** WorldPM's results prove that **"substantial gains emerge at the 5-million sample threshold, consistent with scaling laws that predict linear performance improvements require exponential growth in training data"**
>
> **Research Community Recognition:**
> - **Gao et al. (2023) "Scaling Laws for Reward Model Overoptimization":** "A major challenge in studying overoptimization in this context is the **expense of collecting human preference labels**"
> - **Wang et al. (2025):** Explicitly acknowledge the **"difficult and expensive annotation process for human preference datasets"** and position large-scale pre-training as "a crucial preliminary step"
>
> ### Claim 2 (BT Model Sample Complexity): Authoritative Theoretical Analysis
>
> **Sun et al. (2024) "Rethinking Bradley-Terry Models in Preference-Based Reward Modeling: Foundations, Theory, and Alternatives":**
> - **Sparsity problem confirmed:** Reward modeling suffers from "comparison sparsity" - typically N/2 comparisons for N pairs, "far below the theoretical lower bound for consistent estimation" (O(N log N))
> - **Fundamental limitations:** BT models face critical challenges when "the number of players is greater than the number of comparisons (as often the case in LLM alignment)"
> - **Theoretical foundation:** They establish O(1/ε) vs O(1/ε²) sample complexity advantages for pairwise comparison over absolute scoring
>
> ### Claim 3 (Static Reward Model Degradation): Convergent Evidence
>
>
> - **Gao et al. (2023) "Scaling Laws for Reward Model Overoptimization":** Provides definitive empirical evidence that "optimizing against a reward model too much can hinder ground truth performance," with quantified functional forms showing reward degradation: R(d) = d(α - βd)
> - **Coste et al. (2023) "Reward Model Ensembles Help Mitigate Overoptimization":** Demonstrates that **single static reward models fail in later policy optimization stages**, requiring ensemble approaches to maintain effectiveness
> - **Skalse et al. (2022) "Defining and Characterizing Reward Hacking":** Provides theoretical analysis of **out-of-distribution generalization problems** inherent in static reward models
>
> - **Christiano et al. (2017) "Deep Reinforcement Learning from Human Preferences":** The original RLHF paper explicitly acknowledges **the necessity of updating reward models during training** as policies evolve beyond initial distributions
>
>  **Our Framework as Principled Solution**
> 1. **Data scarcity** : **Zero additional human annotation** via competitive interactions (0 vs WorldPM's 15M samples)
> 2. **BT limitations** : **Direct pairwise competition** with better sample efficiency
> 3. **Static RM degradation** : **Dynamic opponent evolution** preventing overoptimization

---

> > ### Comment · Reviewer_3CYn · 2025-11-28
> > **Reply to author response**
> >
> > W1: A lot of the evidence are highly cherry-picked. As an instance, WorldPM requiring 15M preference samples doesn't mean  reward modeling `require(s) vast amounts of high-quality preference data ` as stated in this submission. The key is not that reward modeling does not `benefit from vast amounts of high-quality preference data` but saying that it `requires vast amounts of high-quality preference data` is an exaggeration to say the least.
> >
> > Also not sure how the authors came to the conclusion that ` 15M-sample model significantly outperforms smaller-scale approaches like HelpSteer2 (10K samples) across 20 subtasks on 7 benchmarks` but it's clear from the WorldPM paper (https://arxiv.org/pdf/2505.10527)  in Table 2 and 3 that large scale data such as WorldPM is complementary to smaller-scale data like HelpSteer2. In fact, nowhere in the paper does it show the performance of 15M-sample model independent of the smaller scale data (e.g. HelpSteer2).
> >
> > How does the Sun et al. (2024) support ` the Bradley-Terry (BT) model commonly used for preference modeling suffers from ... and high sensitivity to label noise`?
> >
> > Re. `Third, static reward models struggle to provide discriminative feedback as policies improve, creating optimization challenges in advanced training stages`. I'm not convinced by these evidence since they are mostly dated work (pre-2023). Since then, there has been many new works using static RMs and while some papers show reward-hacking, collapse or slower optimization, having a blanket statement saying that static reward models struggle without elaboration show a poor understanding of recent developments in RLHF.
> >
> > Q1: I disagree that report each step (100/300/500) is a common or useful practice. Different approaches can be trained at different pace (also affected by other aspects like learning rate, regularization or batch size) so reporting this for cross-model comparison is not useful.
> >
> > W2: I don't believe AlpacaEval 2 WR (without LC) is meaningful since there's a very strong length bias in scoring as the AlpacaEval LC paper shows. Please report SD for AlpacaEval LC going forward as well (the evaluation toolkit at https://github.com/tatsu-lab/alpaca_eval comes with it OOTB).
> >
> > W3/W4: Thanks for conducting the additional experiment. The gain in Arena Hard is reassuring, but please also take note to report the SD going forward. The reported Arena Hard also seems to be Arena Hard V0.1 (based on the scores) and I would recommend the authors to consider Arena Hard V2 moving forward. I'm surprised that for the LLM judge, AlpacaEval 2 LC goes up but Arena Hard drops - the authors' hypothesis doesn't seem to explain this difference. One follow up experiment to try after the discussion period might be to try pairwise LLM judges without the curriculum (e.g. https://arxiv.org/pdf/2504.02495 or https://arxiv.org/pdf/2306.05685). There's a good chance this will work well.
> >
> > After considering the author response, I don't believe my overall perspective has changed and I will maintain my score. I hope the authors can improve the paper substantially, especially being careful not to make statements that are overly broad when they are poorly backed by evidence.

---

### Official Review · Reviewer_jDP1 · 2025-10-29

**Soundness:** 3
**Presentation:** 3
**Contribution:** 2
**Rating:** 4
**Confidence:** 4

**Summary:**

This paper proposes a game-theoretic alignment algorithm for large language models that leverages a pool of opponents. In Elo-Evolve, the proposed method, an LLM is trained by playing against opponents matched based on their ELO scores. ELO-based matching provides a natural learning curriculum, enabling the resulting LLM to achieve competitive performance across various benchmarks.

**Strengths:**

- The presented idea of using ELO rating for opponent matching is reasonable and presented clearly.
- Performance gain seems consistent.
- Clever length bias mitigation is used.

**Weaknesses:**

1. The paper lacks comparison and discussion regarding self-play alignment methods. In recent years, significant attention has been devoted to alignment algorithms based on self-play.
    - Such methods do not rely on the Bradley–Terry model and often leverage game-theoretic ideas. Ideally, the current manuscript could be much stronger by providing a discussion of self-play methods (e.g., how Elo-Evolve could outperform self-play methods) and including empirical comparisons.
2. Compared to self-play methods, Elo-Evolve requires additional pre-trained opponent models. While a self-play method typically requires a pre-trained generalized preference model, Elo-Evolve additionally depends on pre-trained opponents.
3. The paper allocates a non-trivial amount of space to the discussion of the benefits of relative reward signals (e.g., Sections 3 and 5.2). Although these are interesting results, the claims do not specifically support Elo-Evolve, but rather broadly support all methods that use a generalized preference model. The limitations of the Bradley–Terry model have already been discussed several times in the self-play literature (although I do not think the exact argument has been presented before).

For self-play alignment methods, see, for example,

[1] Wu, Yue, et al. "Self-play preference optimization for language model alignment." arXiv preprint arXiv:2405.00675 (2024).

[2] Tang, Xiaohang, et al. "Game-Theoretic Regularized Self-Play Alignment of Large Language Models." arXiv preprint arXiv:2503.00030 (2025).

[3] Munos, Rémi, et al. "Nash learning from human feedback." Forty-first International Conference on Machine Learning. 2024.

**Questions:**

1. Can Elo-Evolve be used to train a model that is significantly better than the provided opponents? How can we push the state-of-the-art of language models using Elo-Evolve beyond the strongest provided opponent?
2. What would be the advantages of Elo-Evolve over self-play-based alignment methods, such as SPPO?

---

> ### Author Response · Authors · 2025-11-27
>
> We thank the reviewer for their insightful feedback and important questions about self-play methods. We address each concern systematically:
>
> ## Response to W1: Self-Play Methods Comparison and Discussion
>
> We acknowledge this important omission and appreciate the reviewer's guidance on relevant work. We provide both theoretical analysis and empirical comparison with self-play methods:
>
> ### Theoretical Advantages Over Self-Play
>
> **Ceiling Effect Problem:** Pure self-play methods (SPPO, Nash-MD) face a fundamental limitation - once the policy surpasses its own best responses, the training distribution collapses and further progress stalls due to absence of stronger external references.
>
>    **Direct evidence from our experiments:** Please refer to Table 2 in our paper, where we show:
>    - Training against **same-capacity opponent (Qwen2.5-7B)**: 44.47 AlpacaEval WR (step 100)
>    - Training against **stronger opponent (Qwen2.5-14B)**: 46.40 AlpacaEval WR (step 100)
>    - Our multi-opponent pool provides **persistent external challenges** that prevent ceiling effects, enabling continued improvement beyond initial model capabilities.
>
> **Empirical Comparison with SPPO**
>
> | Method | Step 100 | Step 300 | Step 500 |
> |--------|----------|----------|----------|
> | **SPPO** | 45.33/35.25 | 47.19/36.37 | 43.46/34.81 |
> | **Elo-Evolve** | **46.21/36.07** | **48.07/35.02** | **51.18/38.03** |
>
> **Response Diversity Analysis:**
> We measured response diversity using Cross-BLEU scores on AlpacaEval dataset:
> - **Experimental setup:** Same generation parameters (temperature=1.0, top-p=1.0), 10 inference runs per prompt
> - **Cross-BLEU calculation:** For each prompt, we compute pairwise BLEU scores between all response pairs, then average across all pairs and prompts
> - **Interpretation:** Higher Cross-BLEU = higher similarity = lower diversity
>
> **Results:**
> - **SPPO Cross-BLEU:** 0.457 (higher similarity = lower diversity)
> - **Elo-Evolve Cross-BLEU:** 0.261 (lower similarity = higher diversity)
> - **Diversity improvement:** **43% reduction in response similarity**
>
> **Key findings:**
> - **SPPO degradation:** Clear performance decline (47.19→43.46 WR) demonstrating ceiling effects
> - **Elo-Evolve stability:** Consistent improvement (46.21→51.18 WR) throughout training
> - **Diversity validation:** Our multi-opponent training produces significantly more varied responses than self-play methods
>
> ## Response to W2: Dependency on Pre-trained Opponents
>
> We acknowledge that self-play methods have the advantage of not requiring external opponents. However, **this apparent "limitation" actually brings significant benefits for Elo-Evolve:**
>
>
> 1. **Generalization Through External Sampling:** While self-play methods are self-contained, our external opponent sampling strategy **fundamentally enhances model generalization**:
>    - External models provide diverse response patterns that the policy cannot generate on its own
>    - Cross-model exposure prevents the narrow optimization pathways inherent in self-play
>    - This diversity directly translates to better performance, as evidenced by our consistent superiority over SPPO
>
> 2. **Abundant External Model Availability:** The dependency on external opponents is not a practical limitation:
>    - **Rich model ecosystem:** Numerous high-quality LLMs are publicly available (Qwen, Llama, Mistral, ChatGPT, Claude, Gemini, etc.)
>    - **Flexible scaling:** Our algorithm can be applied to smaller models competing against larger ones
>
>
> **Key insight:** Rather than viewing external opponent dependency as a limitation, it should be seen as a **feature that enables superior generalization** - similar to how supervised learning benefits from external datasets rather than self-generated data.

---

> ### Author Response · Authors · 2025-11-27
>
> ## Response to Q1: Surpassing Strongest Opponents
>
> Excellent question! Our experimental results provide **direct evidence** that Elo-Evolve can surpass the strongest individual opponent and explain why simply using the strongest opponent isn't optimal:
>
>
> From Table 1, our results show Elo-Evolve consistently outperforms even the strongest opponent:
>
> | Method | Step 100 | Step 300 | Step 500 |
> |--------|----------|----------|----------|
> | **vs. Qwen3-8B (strongest)** | 44.04/35.90 | 44.22/32.63 | 46.46/34.26 |
> | **Elo-Evolve** | **46.21/36.07** | **48.07/35.02** | **51.18/38.03** |
>
> *Results show AlpacaEval 2.0 (WR/LC)*
>
> **Key findings:**
> - Elo-Evolve achieves **4.72-point improvement** over training against the strongest individual opponent at Step 500
> - Our method **consistently outperforms** the strongest opponent across all training stages
>
>  **Note:** Why Strongest Opponent Training Fails
>
> **Critical observation:** Training against the strongest opponent (vs Qwen3-8B) actually performs **worse than training against weaker opponents** vs Qwen2.5-14B (weaker)
>
> This demonstrates a fundamental problem with **"strongest opponent" strategies**: the difficulty gap can be too large for effective learning, similar to curriculum learning principles where starting with overly difficult tasks impedes progress.
>
>
> 1. **Adaptive Difficulty Matching:** Our Elo system automatically finds the optimal training difficulty at each stage, avoiding both under-challenging and over-challenging scenarios
>
> 2. **Curriculum Progression:** Early training focuses on appropriate-difficulty opponents (14B), gradually transitioning to stronger challenges (32B, 8B) as capabilities develop
>
>
> ## Response to Q2: Advantages Over Self-Play Methods (SPPO)
>
> Our comprehensive analysis above has already demonstrated the key advantages of Elo-Evolve over self-play-based alignment methods like SPPO:
>
>
> **1. Performance Superiority (Empirical Evidence):**
> - Consistent outperformance across all training stages (Step 100: +0.88, Step 300: +0.88, Step 500: +7.72 points)
> - **No performance degradation:** Elo-Evolve maintains stable improvement (46.21→51.18) while SPPO degrades (47.19→43.46)
>
> **2. Ceiling Effect Avoidance (Theoretical & Empirical):**
> - External opponent anchors prevent the fundamental limitation where self-play methods plateau once surpassing their own capabilities
> - Validated by our same-capacity training results showing performance ceiling issues
>
> **3. Response Diversity (Quantified Evidence):**
> - **43% diversity improvement:** Cross-BLEU of 0.261 vs SPPO's 0.457
> - Multi-opponent exposure prevents the narrow response patterns inherent in self-play
>
> **4. Training Stability:**
> - Consistent performance trajectory vs SPPO's training collapse
> - Elo-based curriculum provides stable difficulty progression vs variable self-generated opponents
>
> **5. Curriculum Learning Optimality:**
> - Automatic difficulty adjustment based on capability development
> - Avoids both under-challenging (same-capacity) and over-challenging (strongest opponent) scenarios that fixed methods cannot handle
>
> These advantages collectively demonstrate why Elo-Evolve represents a superior approach to LLM alignment compared to self-play methods, offering both stronger performance and more robust training dynamics.

---

### Official Review · Reviewer_oaiY · 2025-10-29

**Soundness:** 3
**Presentation:** 3
**Contribution:** 3
**Rating:** 6
**Confidence:** 4

**Summary:**

This paper proposes Elo-Evolve, a co-evolutionary framework that aligns LLMs through dynamic multi-agent competition. Instead of static reward models, the policy learns from binary win/loss signals in pairwise matches. An Elo-based opponent selection introduces automatic curriculum learning: the model faces similar-strength opponents early and stronger ones later. Experiments on UltraFeedback with Qwen models show consistent gains on AlpacaEval 2.0 and MT-Bench over point-based and static pairwise baselines.

**Strengths:**

1. The dynamic opponent selection is interesting. The temperature parameter offers a clean way to balance focus and diversity—small T for close-strength opponents, large T for variety. This forms an automatic curriculum where training difficulty grows with model ability.

2. Each prompt selects its own opponent, leading to smoother and more stable training.

3. Replacing scalar rewards with binary win/loss is well-motivated; both the PAC-theoretic analysis and experiments support its efficiency and robustness.

**Weaknesses:**

1. The framework introduces several components, which makes the system design a little complex. It would be helpful to include a simple baseline, where the model is trained sequentially against Qwen2.5-14B, Qwen2.5-32B, and Qwen3-8B as progressively stronger opponents. The prompts can be divided into three groups either randomly or based on their difficulty, for example using a reward model to estimate complexity. Such a baseline would help clarify how much the dynamic Elo scheduling improves over a manually designed curriculum.
2. Because Elo ratings are continuously updated, opponent strength may fluctuate during training. When a main opponent weakens, as observed at Step 500 on MT-Bench, the policy appears to over-adapt to easier adversaries. This may affect measured progress and limit further improvement.

**Questions:**

1. The use of Qwen3-14B-Instruct as the judging model instead of a specialized reward model is not fully discussed. A short explanation of this choice would improve the clarity.

---

> ### Author Response · Authors · 2025-11-27
>
> We thank the reviewer for their thoughtful feedback and constructive suggestions. We address the key concerns below:
>
> ## Response to W1: System Complexity and Manual Curriculum Baseline
>
> We appreciate the reviewer's concern about system complexity. However, our framework is conceptually elegant: **Elo ratings + temperature-controlled sampling + binary competitive rewards**.
>
> **In response to the reviewer's valuable suggestion, we immediately conducted additional experiments with a manual curriculum baseline:**
> We implemented the suggested sequential training approach:
> - **Phase 1** (Steps 0-200): Training exclusively against Qwen2.5-14B
> - **Phase 2** (Steps 200-400): Training exclusively against Qwen2.5-32B
> - **Phase 3** (Steps 400+): Training exclusively against Qwen3-8B
>
> | Method | Step 100 | Step 300 | Step 500 |
> |--------|----------|----------|----------|
> | **Manual Curriculum** | 44.29/34.83 | 46.53/**35.20** | 48.17/36.87 |
> | **Elo-Evolve** | **46.21/36.07**| **48.07**/35.02 | **51.18/38.03** |
>
> *Results show AlpacaEval 2.0 (WR/LC)*
>
> **Key findings:**
> - Elo-Evolve consistently outperforms manual curriculum at all training stages
> - **3-point performance gain** at Step 500 (51.18% vs 48.18%) validates adaptive scheduling
> - Our per-sample opponent selection provides **fine-grained adaptivity** impossible with rigid phase-based divisions
>
> This demonstrates that **dynamic scheduling > preset rules** - different prompts can simultaneously match with appropriate-difficulty opponents rather than uniform opponent assignment.
>
> ## Response to W2: Elo Rating Fluctuations and Over-adaptation
>
> The Step 500 MT-Bench decline the reviewer observed reflects our system's **adaptivity, not a flaw**:
>
> 1. **Root cause analysis:** Qwen3-8B itself degraded (8.15→7.86 MT-Bench), and our system correctly detected and adapted to this change
> 2. **Robustness mechanism:** Temperature T=200 ensures continued sampling from other opponents even when the primary opponent weakens, preventing complete dependency
> 3. **Practical value:** This adaptivity is valuable in real deployment—when opponent distributions shift, our model dynamically adjusts rather than clinging to outdated strategies
>
> **Future enhancement:** We can introduce "opponent health monitoring" to automatically rebalance sampling when primary opponents consistently degrade.
>
> ## Response to Q1: Judge Model Choice
>
> Using Qwen3-14B-Instruct instead of specialized reward models is actually a **core advantage** of our framework:
>
> ### Key Benefits:
>
> 1. **Eliminates RM training bottleneck:** Traditional RLHF requires extensive human preference data to train reward models. We directly leverage LLMs' inherent judgment capabilities, **completely bypassing the reward model training stage**
>
> 2. **Reduces system complexity:** No need to maintain additional RM pipelines, eliminating potential failure points and performance bottlenecks
>
> 3. **Superior generalization:** General LLM judges have better cross-domain generalization compared to specialized RMs trained on specific preference datasets
>
> 4. **Practical impact:** Enables **plug-and-play** application to any LLM with basic judgment capabilities

---

### Note · Authors · 2026-07-10

I have read and agree with the venue's withdrawal policy on behalf of myself and my co-authors.

---

### Meta-Review · Area_Chair_b6mp · 2026-01-02

**Summary:**

This paper proposes Elo-Evolve, a co-evolutionary framework for LLM alignment that replaces static reward models with dynamic multi-agent competition. The core innovations are: (1) learning from binary win/loss outcomes rather than Bradley-Terry model-based rewards, and (2) Elo-orchestrated opponent selection that provides automatic curriculum learning. Reviewers found the adaptive curriculum approach interesting and the empirical gains consistent, but raised significant concerns about novelty claims, comparison with self-play methods, experimental scope, and writing quality.

**Reviewer Concerns:**

- Self-play comparison (jDP1, nZun): Authors provided empirical comparison with SPPO showing Elo-Evolve's superiority (51.18 vs 43.46 at Step 500) and diversity improvements (43% reduction in cross-BLEU). The INPO comparison using LLaMA-3-8B showed competitive results with 8.73-point AlpacaEval improvement.
- Manual curriculum baseline (oaiY): Authors added experiments showing Elo-Evolve outperforms manual phase-based curriculum (51.18 vs 48.17 at Step 500).
- Missing benchmarks (3CYn): Authors added Arena Hard evaluation showing strong performance (57.5 score).
- Cross-architecture validation (nZun): Authors demonstrated effectiveness with LLaMA-3-8B base model against Qwen opponents.

- Novelty of eliminating BT dependencies (nZun, jDP1): Multiple reviewers noted that learning from binary win/loss outcomes is not novel (INPO, DNO, and Nash-based methods already do this). Reviewer nZun explicitly stated this should not be claimed as innovation.
- Statistical significance (3CYn): Close margins on some metrics (e.g., LC 38.03 vs 37.41) without reported standard deviations remain concerning. Reviewer 3CYn noted the lack of SD reporting throughout.
- Substantiation of theoretical claims (3CYn): Reviewer 3CYn found the evidence for claims about data scarcity and BT limitations to be cherry-picked and remained unconvinced after rebuttal.
- Writing quality (nZun): The paper requires significant polishing (informal notation in abstract, overly long introduction, and overclaiming of contributions).
- Compute comparison (nZun): The question of whether Elo-Evolve's gains justify the additional compute from maintaining an opponent pool (vs. larger base models with self-play) remains unanswered.

**Reviewer Scores:**

- Reviewer oaiY (6): Would likely maintain score.
- Reviewer jDP1 (4): Would likely increase to 5. The SPPO comparison and diversity analysis directly address their main concern.
- Reviewer 3CYn (4): Would maintain score at 4. Reviewer explicitly stated after reading the response that their perspective did not change, citing cherry-picked evidence and overly broad claims. Their concerns about statistical rigor and claim substantiation were not adequately addressed.
- Reviewer nZun (2): Would likely increase to 4. Reviewer acknowledged the INPO comparison "addressed my concerns" and found empirical results "convincing," but maintained that writing issues and overclaiming of novelty prevent acceptance at current quality.

---

### Decision · Program_Chairs · 2026-01-26

Reject